# A Deep Learning-Based Automated Framework for Subpeak Designation on Intracranial Pressure Signals

**DOI:** 10.3390/s23187834

**Published:** 2023-09-12

**Authors:** Donatien Legé, Laurent Gergelé, Marion Prud’homme, Jean-Christophe Lapayre, Yoann Launey, Julien Henriet

**Affiliations:** 1Sophysa, 91400 Orsay, France; mprudhomme@sophysa.com; 2DISC Department, FEMTO-ST, Université de Franche-Comté, 25000 Besançon, France; jean-christophe.lapayre@femto-st.fr (J.-C.L.); julien.henriet@femto-st.fr (J.H.); 3Intensive Care Unit, University Hospital of Saint-Etienne, 42100 Saint-Etienne, France; laurentgergele@yahoo.fr; 4Intensive Care Unit, University Hospital of Rennes, 35000 Rennes, France; yoann.launey@chu-rennes.fr

**Keywords:** intracranial pressure, cerebral compliance, deep neural networks, recurrent neural networks, convolutional neural networks

## Abstract

The intracranial pressure (ICP) signal, as monitored on patients in intensive care units, contains pulses of cardiac origin, where P1 and P2 subpeaks can often be observed. When calculable, the ratio of their relative amplitudes is an indicator of the patient’s cerebral compliance. This characterization is particularly informative for the overall state of the cerebrospinal system. The aim of this study is to develop and assess the performances of a deep learning-based pipeline for P2/P1 ratio computation that only takes a raw ICP signal as an input. The output P2/P1 ratio signal can be discontinuous since P1 and P2 subpeaks are not always visible. The proposed pipeline performs four tasks, namely (i) heartbeat-induced pulse detection, (ii) pulse selection, (iii) P1 and P2 designation, and (iv) signal smoothing and outlier removal. For tasks (i) and (ii), the performance of a recurrent neural network is compared to that of a convolutional neural network. The final algorithm is evaluated on a 4344-pulse testing dataset sampled from 10 patient recordings. Pulse selection is achieved with an area under the curve of 0.90, whereas the subpeak designation algorithm identifies pulses with a P2/P1 ratio > 1 with 97.3% accuracy. Although it still needs to be evaluated on a larger number of labeled recordings, our automated P2/P1 ratio calculation framework appears to be a promising tool that can be easily embedded into bedside monitoring devices.

## 1. Introduction

Intracranial pressure (ICP) is classically monitored invasively in intensive care units (ICU) in the event of brain injury. One of the main objectives for a clinician is to limit the time the patient remains above a threshold of cerebral hypertension, as described by international guidelines [1]. Beyond the analysis of the mean ICP, the ICP signal is a combination of different periodic components, which are affected by cardiac and respiratory frequencies. Thus, the sole mean ICP cannot capture all the information provided by such a complex signal [2]. For instance, this single number does not describe the ability of the cerebrospinal system to compensate for the changes in volume caused by blood and cerebrospinal fluid (CSF) displacements so that the ICP is maintained in an acceptable range. This pressure–volume relationship, generally called “cerebral compliance” [3], requires fastidious manipulations to be measured punctually with CSF infusion tests [4,5,6]. Combined with the absolute value of the ICP, a measure of cerebral compliance could help guide the clinical management of intensive care patients [7,8,9].

This is why different characterizations of cerebral compliance, based on a mathematical analysis of the ICP waveform, have been proposed in the literature [10,11]. Notably, the shape of heartbeat-induced pulses varies according to cerebral compliance [12]. When the latter is at a normal state, three subpeaks of decreasing amplitudes are generally visible (see Figure 1). These peaks are called P1, P2, and P3, in accordance with their apparitional order. While it is broadly accepted that P1 is due to the systolic pressure wave, the origins of P2 and P3 remain unclear [13]. MRI measurements tend to associate P2 with the maximum volume in the cerebral arteries [14,15], whereas P3, classically described as the dicrotic wave, could be linked to the veinous outflow [16]. In any case, as cerebral compliance decreases, P2 and P3 become increasingly higher compared to P1 [17]. At the same time, their appearance times become closer [18] until the pulse takes a triangular shape centered on P2. Therefore, the ratio of the relative amplitudes of P2 and P1 (designated as the P2/P1 ratio) has been used as an indicator of cerebral compliance [19]. This ratio is all the more relevant given that Kazimierska et al. [20] demonstrated its good correlation with cerebral compliance assessed by standard infusion tests. As a consequence, from an end-user point of view, a P2/P1 ratio inferior to 1 can be interpreted as a sign of good cerebral compliance. However, the underlying physiological mechanisms at the origin of the changes in the relative heights of the subpeaks are still debated. Recent mechanistic modeling supported by MRI measurements suggests that multiple parameters can influence the final ICP pulse shape [21]. Notably, a low elastance of the cerebral arteries and/or a high elastance of the cerebral veins seem to be associated with pulse configurations where P2 is higher than P1.

From a practical perspective, the automated detection of P1 and P2 on an ICP signal faces different issues due to the highly variable pulse morphology. Only a few automated frameworks allowing for P2 and P1 designation have been proposed in the literature [22,23,24,25]. Most of them rely on clustering algorithms to only analyze one characteristic pulse over a predefined period, as proposed by the authors of the Morphological Clustering and Analysis of Continuous Intracranial Pressure (MOCAIP) algorithm [26]. MOCAIP-based automated frameworks are designed to compute a large number of morphological features of the ICP pulses, including the P2/P1 ratio. However, in addition to the raw ICP signal, their data processing workflows require both electrocardiogram (ECG) monitoring and an extensive reference library of non-artifact pulses, which can be difficult to implement into an onboard bedside device. To perform real-time P2/P1 ratio calculations, neural network-based algorithms seem to be the tool of choice to circumvent these prerequisites due to their ability to directly integrate the information provided by previous examples into trained models. For instance, convolutional neural networks (CNN) and long short-term memory (LSTM) recurrent networks have been used successfully for similar tasks, such as ECG beat detection and classification (respectively, [27,28,29,30,31]).

Under the constraint of only using an ICP signal as an input, we developed a deep learning-based framework to detect the P2 and P1 subpeaks and compute the ratio of their relative amplitudes where possible. Its conception was performed by achieving a comparative study of the proposed deep learning network architectures, enhanced with pre- and post-treatments, and applied to our dataset provided by the ICU at the University Hospital of Saint-Etienne. Our framework is designed to perform two tasks sequentially. The first is a classification task that aims to eliminate all pulses without P1 and P2 subpeaks. The second, which is only performed on the remaining pulses, aims to identify the P1 and P2 subpeaks to calculate the ratio of their relative amplitudes. As an output, our framework provides a discontinuous signal of P2/P1 ratio values, postprocessed to make it as readable as possible for the clinician. In this article, we provide a description of the neural network (NN) architectures we compared for pulse selection (Section 3.2) and subpeak designation (Section 3.3). The performance obtained for each of these tasks is reported in Section 4.1 and Section 4.2, respectively. Lastly, we test our completed automated framework on a dedicated testing dataset (Section 4.3).

## 2. Dataset Overview

The studied ICP signals came from 10 adult patients suffering from traumatic brain injury, who were admitted to the ICU of the University Hospital of Saint-Etienne (France) between March 2022 and March 2023. For each of them, the ICP was monitored invasively with an intraparenchymal sensor (Pressio, Sophysa, Orsay, France) for a duration of 8.3 ± 5 days (min = 3.8, max = 15) at a sampling frequency of 100 Hz.

The dataset used in this study to train and select the best-performing NN architectures was created by randomly sampling five one-hour sections from each record. Four of them were allocated to the training dataset, whereas the last one was allocated to the testing dataset. After the pulses were preprocessed and individualized, as described in Section 3.1, 1 out of 15 was selected for inclusion in the final datasets. These pulses were labeled with the positions of P1 and P2 if both of them were visible, or [0, 0] otherwise. In the end, the training dataset was composed of 13,127 pulses, including 12,308 with a calculable P2/P1 ratio. Its testing counterpart was composed of 4344 pulses, including 3847 with a calculable P2/P1 ratio. These proportions are in accordance with Rashidinejad et al., who estimated a missing subpeak probability of less than 10% based on their 700-h dataset [25].

To assess the performance of the final dataset under more realistic conditions, an additional 10-min segment was sampled randomly from each of the 10 patients. This second testing dataset, divided into 10 contiguous segments, was composed of 7399 pulses, of which 6815 had a calculable P2/P1 ratio. The first dataset was designed to capture maximum diversity among the patients, whereas the second one was designed to assess the performance of the entire automated framework on continuous recordings.

## 3. Materials and Methods

Our data processing pipeline is divided into four parts. After a heartbeat-induced pulse detection step performed on a preprocessed ICP signal, artifacts and pulses without a calculable P2/P1 ratio are eliminated using a deep learning-based algorithm. The subpeaks are then detected on the remaining pulses. Lastly, a postprocessing step is performed to remove outliers and deal with missing values.

### 3.1. Data Preprocessing

A fourth-order Butterworth bandpass filter between 0.3 Hz and 20 Hz is first applied to the raw signal. It is meant to isolate cardiac pulses from rapid oscillations of electronic origin, respiratory waves, and baseline variations. The modified Scholkmann algorithm is then applied to the filtered signal to detect the pulse onsets [32]. As the patients’ pulse rates range between approximately 60 and 80 bpm, the characteristic duration L provided to the algorithm is set at 500 ms. Indeed, this hyperparameter is supposed to represent at least a quarter of the average pulse duration. The amplitude of each single pulse is normalized between 0 and 1, whereas the length is set to 180 points through a third-degree polynomial interpolation. This preprocessing step is nearly identical to the one performed by Mataczynski et al. for pulse-shape index calculations [33], except for the filter applied to the raw signal. As an output, an *N* × 180 matrix of *N* pulses is provided to the selection algorithm. Two examples of preprocessed signals are shown in Figure 2.

### 3.2. Pulse Selection

A major difficulty in monitoring the P2/P1 ratio is that not all subpeaks are systematically visible on all pulses. Therefore, a selection step is needed so that the detection algorithm is only provided for pulses where P1 and P2 are visible. This selection is performed by a neural network. Three architectures are compared for this task, namely a one-dimensional CNN, an LSTM-based recurrent network, and a long short-term memory fully convolutional network (LSTM-FCN), which is a combination of both. All the models are trained to perform the same binary classification task by minimizing a Binary Cross-Entropy (BCE) loss. Before calculating the loss function, a sigmoid is applied to the neural network outputs to obtain values between 0 and 1.

#### 3.2.1. One-Dimensional CNN Architecture

These architectures extract relevant features by applying convolutional filters to the input tensor. CNNs have been successfully used for medical image segmentation, but it is also possible to adapt the layers’ dimensions to process one-dimensional vectors in the same way. Our CNN comprises three encoding blocks, each composed of the sequence convolutional layer–batch normalization–ReLU activation, followed by a max pooling layer. The output is postprocessed by two dense layers separated by a ReLU activation layer. To reduce overfitting, a dropout with a probability of 0.2 is applied at the end of the encoder and to the first dense layer. The dimensions of each layer are shown in Figure 3.

#### 3.2.2. LSTM-Based Recurrent Network

Recurrent networks are designed to capture the underlying time dependencies of sequential data. They are generally composed of one or more cells whose outputs are computed based on the current input state and the outputs of previous states. Past predictions can be taken into account in different ways; LSTM cells are specifically designed to track long-term dependencies [34]. The proposed recurrent network is a single bi-directional LSTM cell, followed by two dense layers separated by a ReLU activation layer. Hence, the input vector is processed in both reading directions by the LSTM cell, which produces two outputs that are concatenated and postprocessed by the two dense layers. A dropout with a probability of 0.2 is applied at the end of the LSTM cell and to the first dense layer. The dimensions of each layer are shown in Figure 3.

#### 3.2.3. LSTM-FCN Network

The two above-mentioned architectures process the input data with different objectives. CNNs focus on the neighborhood of each point, whereas recurrent neural networks are used to exploit the causalities inherent in sequential data. LSTM-FCN networks attempt to combine both strategies and were specifically designed for time-series classification [35]. Moreover, Mataczynski et al. obtained good results with such an architecture for pulse-shape index calculations [33]. The LSTM-FCN network we implement contains a three-block encoder, placed in parallel with an LSTM cell. Their respective dimensions are identical to those used for the CNN and the LSTM-based recurrent network. Both the computations are performed in parallel. The outputs are then concatenated and processed by two dense layers. As above, a dropout with a probability of 0.2 is applied to the first dense layer.

### 3.3. Subpeak Designation

Once the pulses with a calculable P2/P1 ratio are selected, the P1 and P2 subpeaks can be designated. The identification strategy is divided into two steps. Firstly, a set of candidate subpeaks is defined using the pulse curvature, as is the case for the MOCAIP-based algorithms [26]. Secondly, two of these candidates are designated as P1 and P2 on the basis of a neural network output.

#### 3.3.1. Candidate Subpeak Detection

The set of candidate subpeaks corresponds to all the local maxima of the pulse curvature κ, which is defined as follows:$$\kappa \left(x\right)=\frac{x^{\prime \prime }}{{\left(1+x^{\prime 2}\right)}^{3/2}}$$

It is assumed that for a given pulse *p*, the P1 and P2 subpeaks correspond to two local minima of κ(p) located in zones where $p^{\prime \prime }$ is negative or, equivalently, to two local maxima of κ(−p).

#### 3.3.2. P1 and P2 Designation

In parallel to these calculations, neural networks learn a classification task. The aim is to assign a value between 0 and 1 to each of the 180 points of the pulse, such that a value close to 1 denotes the proximity of a P1 or P2 subpeak. More formally, for a pulsation *x*, the objective of the classification task is a 180-point vector $y_{x}$, such that
$$\forall t\in \left[\;\left[1,180\right]\;\right],y_{x}\left(t\right)=\frac{1}{2}\left(e^{\frac{-{\left(x\left(t\right)-p_{1}\left(x\right)\right)}^{2}}{2}}+e^{\frac{-{\left(x\left(t\right)-p_{2}\left(x\right)\right)}^{2}}{2}}\right)$$
where $p_{1}\left(x\right)$ and $p_{2}\left(x\right)$ are the respective positions of P1 and P2. During the learning process, the neural networks seek a function $f^{*}$, such that
$$f^{*}=argmin_{f}\sum_{x\in D}MSE\left(f\left(x\right),y_{x}\right)$$
where MSE denotes the *Mean Square Error* loss function, and *D* is the training set. The two candidates that correspond to the two highest values of ${\hat{f}}^{*}$ are designated as P1 and P2 according to their order of appearance.

Two network architectures are compared for the estimation of $f^{*}$, namely a 1-dimensional U-Net (see Section 3.3.3) and an LSTM-based recurrent network (see Section 3.3.4). For completeness, during the experiments, the designation of P1 and P2 is also performed on the basis of the two highest local maxima of the NN output. Both strategies are summarized in Figure 4. As a result, four designation methods are compared, namely the U-Net and the LSTM network with and without the candidate selection step.

#### 3.3.3. One-Dimensional U-Net

U-Net is a type of CNN architecture. Its three-level bottleneck structure is composed of two symmetric blocks. In addition to the linear information propagations, pairwise connections are set between same-shape components. As it was originally conceived for image segmentation, the layers are modified here to perform one-dimensional convolutions. The layer dimensions are shown in Figure 5. A dropout with a probability of 0.2 is applied to each convolution block.

#### 3.3.4. LSTM-Based Recurrent Network

We use a bidirectional LSTM-based recurrent network similar to the one trained for peak selection (see Section 3.2.2). Hence, the input 180-sample pulse is processed by a single LSTM cell, followed by two consecutive dense layers. As the hidden layer size of the LSTM cell is set to 180, the dimensions of the two consecutive dense layers are set to (360, 360) and (360, 180), respectively. A dropout with a probability of 0.2 is applied to the first dense layer.

### 3.4. Postprocessing

Postprocessing the P2/P1 ratio signal has to address three main issues:Spurious oscillations, mostly due to the intrinsic variability of the ICP signal. Even if they are not a result of the data processing pipeline itself, they tend to make the record less readable for the clinician.Missing values, since all the pulses that do not pass the selection cut are recorded as missing.Punctual outliers. If they are not caused by the ICP signal itself, they can be due to errors in the data processing pipeline. Punctual outliers occur either at the classification step when false-positive pulses are provided to the detection algorithm, or at the detection step when P1 and P2 are designated at incorrect positions.

These different problems are alleviated in the postprocessing phase by retrospectively smoothing the ratio monitoring. To do so, a 95% normal confidence interval is estimated on a 100-pulse sliding window. A mean ratio is then calculated over the window if at least 50 values are non-missing; otherwise, the value corresponding to this window is reported as missing. Therefore, each displayed value is calculated on the basis of the last 100 pulses, which corresponds to about one minute. In addition to overcoming the three issues listed above, smoothing the output signal in such a way highly enhances its readability. Indeed, far too many factors can influence a single pulse P2/P1 ratio to draw any conclusion on the basis of a pulse-wise evolution.

## 4. Results

Experiments were performed separately for the pulse selection and peak detection tasks to select a single neural network for each of them. The same training and testing datasets of labeled pre-processed pulses were used for both tasks, with 10% of the training set used for validation. Upon completing our framework with two trained neural networks, we fully processed 10-min labeled segments randomly sampled from each of the recordings. To ensure the reproducibility of our experiments, each of the three steps was performed using a dedicated processing pipeline designed with Snakemake 7.25 [36]. All the associated scripts were coded in Python 3.11. The neural networks were implemented with Pytorch 2.0 [37]. All the experiments described below were performed on a Windows 10 machine powered by WSL2 Ubuntu 20.04.5, equipped with a 12th Gen Intel(R) Core(TM) i7-12850HX 2.10 GHz 16 CPU, an Nvidia RTX A3000 12GB Laptop GPU, and 16 GB of RAM. The pipelines used for comparing the performance of the neural networks are available at the following address: https://github.com/donatien-lege/P1_P2_detection_ratio, (accessed on 7 September 2023).

### 4.1. Pulse Selection

The three models (i.e., CNN, LSTM recurrent network, and LSTM-FCN) were trained on 150 epochs with the Adam optimizer, an initial learning rate of 0.001, and a batch size of 256. For each of them, the area under the receiver operating characteristic (ROC) curve was calculated by plotting the true-positive rate (TPR) against the false-positive rate (FPR), defined as:$$TPR=\frac{\mathrm{True}\;\mathrm{Positive}}{\mathrm{True}\;\mathrm{Positive}+\mathrm{False}\;\mathrm{Negative}},\;FPR=\frac{\mathrm{False}\;\mathrm{Positive}}{\mathrm{False}\;\mathrm{Positive}+\mathrm{True}\;\mathrm{Negative}}$$

The three ROC curves and the associated precision–recall curves are displayed in Figure 6. For the final framework, the optimal decision threshold was chosen to maximize the difference between the TPR and the FPR (TPR−FPR). The 95%-confidence intervals were calculated using the R package pROC [38].

Our LSTM-based recurrent network architecture outperformed the convolution-based ones, with an area under the curve of 0.905. The confusion matrices corresponding to the respective optimal decision thresholds of each NN architecture are presented in Table 1.

The percentages of false-positive pulses and false-negative pulses amounted to 1.8% and 9.7%, respectively, of the total dataset when using the LSTM-based architecture for classification. In contrast, these percentages amounted to 2.3% and 42.9%, respectively, when using the convolutional neural network. As all the pulses that passed this selection step were provided to the pulse designation algorithm, the high false-positive rate may have resulted in the misleading P2/P1 values finally displayed. On the other hand, the high false-negative rate resulted in a loss of information for the end user since a lot of pulses with a calculable P2/P1 were then discarded by the algorithm.

### 4.2. Peak Designation

The experimental pipeline was designed to compare the four possible combinations between the peak designation method (i.e., by using the curvature function or not) and the neural network architecture (i.e., 1D convolutional U-Net or LSTM-based recurrent network). In addition, a designation using only the first two local maxima of the curvature was performed as a baseline. Both models were trained on 150 epochs with the Adam optimizer, with an initial learning rate of 0.001 and a batch size of 256. The mean absolute peak appearance time error and mean absolute P2/P1 ratio error were calculated. The mean absolute appearance time error was expressed as a percentage of the whole pulse duration. The results are reported in Table 2. In addition, as it is the most interpretable information for a clinician, we assessed the ability of our models to detect pulses where P2 is higher than P1. To do so, we calculated a confusion matrix for classes “+”: “P2/P1 ratio > 1” and “−”: “P2/P1 ratio < 1” and the associated accuracies, defined as the proportion of correct predictions over the whole testing dataset.

As for the pulse selection task, the recurrent architecture outperformed the convolutional one. Without the curvature-based candidate peak selection step, the LSTM-RE architecture performed the classification task with a 3% higher accuracy than our 1D U-Net. Moreover, it achieved the most accurate estimation of the P2/P1 ratio, with a mean average error of 0.03. Performing the selection of the candidate peaks using the means of the curvature function tended to improve the algorithm’s ability to discriminate pulses with a P2/P1 ratio > 1 at the cost of a slightly less accurate ratio estimation.

### 4.3. Final Automated Framework

On the basis of the previous experiments, we finally chose an LSTM-based recurrent network for both the pulse selection and subpeak designation steps. For the latter step, P1 and P2 designation was performed by selecting the two best LSTM-scored local maxima of the curvature. For each of the ten patients, the complete workflow was used to process a randomly chosen labeled 10-min section. An example of such an output is presented in Figure 7. It can be seen that the outliers were effectively discarded by the postprocessing step, as was the case for the trough at 0.10 around the 410th pulse. However, due to the parameters of the sliding window computation, a delay was introduced between the raw P2/P1 ratio signal and the postprocessed output.

The performance of the entire P2/P1 calculation pipeline was assessed for each individual 10-min segment. We used the same metrics as above to assess the pulse selection and subpeak designation tasks. In addition, we calculated the percentage of pulses that were assigned a ratio value and the percentage of non-missing values in the final postprocessed ratio signal. Table 3 contains the values calculated over the total 100-min dataset, and the 10-min individualized segment metrics are available in Table A1.

The false-positive and true-positive rates were both about seven points higher than their respective equivalents calculated when using the NN architecture. However, the subpeak designation performance was consistent with that of previous experiments. Table 4 shows the overall confusion matrix calculated for the pulse selection. As above, the individualized confusion matrices are available in Table A2.

It is noteworthy that only the second segment sample contained 91% of the negatively labeled pulses. In this segment, the pulse selection algorithm was performed with a 13.5% false-positive rate (Table A2).

False-positive pulses and false-negative pulses amounted to 1.14% and 7.49%, respectively, of the total testing dataset. These proportions are consistent with those previously achieved on the 4344-pulse testing dataset.

## 5. Discussion

Our deep learning-based framework was designed to perform P1 and P2 detection and P2/P1 ratio computation directly on a bedside device. For convenience, we designed it under the constraint of only using the ICP signal, which was made possible through a well-established efficient preprocessing step. Hence, we were able to focus our deep learning-based analysis on short time series corresponding to single pulses of cardiac origin. This strategy enabled us to use network architectures that were not too deep. Moreover, working at the cardiac cycle scale allowed us to alleviate another real-life difficulty: in bedside monitoring, ICP signals are very often contaminated with artifacts, either due to patient movements (coughing, reactions to drug administration, nursing manipulations, etc.) or electronic perturbations [39,40]. Therefore, it can be complicated at a macroscopic scale to determine whether an acute rise in the ICP corresponds to a real physiological measurement or artifacts [41]. By only focusing on modified Scholkmann algorithm-extracted candidate pulses, we were able to perform this artifact removal step on the basis of the local waveform alone in the pulse selection step. In addition, as changes in cerebral compliance generally occur in a progressive way [42], a continuous pulse-wise compliance score is the tool of choice to describe the current patient’s state as faithfully as possible.

When labeling the pulses, only using the ICP signal sometimes caused difficulties in interpreting isolated single-pulse waveforms; without other elements of context, pulses with only two visible subpeaks systematically fell into the “non-calculable P2/P1 ratio” category since it was not possible to know whether P1, P2, or P3 was missing. In some of these cases, ABP or ECG signals may have helped to distinguish the subpeaks [20], and thus compute the P2/P1 ratio. In this sense, the training dataset was labeled in a quite restrictive way to limit, as much as possible, the number of pulses without a calculable P2/P1 ratio provided to the peak designation step. However, this decision resulted in inevitable consequences for the amount of time during which a P2/P1 ratio could be displayed. In any case, the recurrent architectures clearly outperformed the convolutional-based ones for pulse selection, even if it was possible to reduce the observed gap by fine-tuning the proposed convolutional architecture. As a full succession of subpeaks is essential for understanding the pulse waveform, our results suggest that recurrent networks may be more suitable than CNNs for performing this classification task. In this sense, these results may be in contrast to those of similar studies performed on ECG signals, where events such as QRS complexes have more recognizable shapes, thus making CNNs relevant for classification or detection tasks [43,44]. Concerning the consequences of misclassified pulses, it was notable that false-negative pulses only caused spurious missing values at the end of the data processing workflow. In contrast, false-positive pulses were provided to the peak designation algorithm that systematically output the two positions of the estimated P1 and P2. Therefore, false-positive pulses can cause much more damage to the output P2/P1 ratio signal. While we chose an optimal threshold that minimized the difference between the true-positive and false-positive rates, it may be possible to change the decision boundary for better overall accuracy. The performance achieved using the chosen threshold corresponded to an accuracy of 78%. Because our primary goal was to use an algorithm that was as restrictive as possible, this result is lower than the results using other pulse classification algorithms, which achieved accuracies between 82% [33] and 88% [45] in a multiple-class problem. Indeed, if false-positive pulses are rare enough, it is then possible to smooth their induced misleading P2/P1 ratio values with artifacts in the postprocessing step (Figure 7).

Peak detection was performed by computing a density function using the neural networks, as is often the case in image segmentation tasks. We chose to stick to the underlying philosophy of MOCAIP-based automated frameworks [26], which include a candidate selection step before subpeak designation. It would have been possible to turn our algorithm into a regression task to output the estimated positions directly, as is sometimes done for the detection of ECG peaks [28]. This simpler strategy would have led to lighter computations. However, our method offers two advantages. Firstly, it is more robust and explainable in itself, as a score is assigned to each point in the input tensor. Secondly, it is easier to combine the output tensor with another function such as the pulse curvature. Designating two peaks from among a set of candidates selected with this simple and explainable criterion offers guarantees for the generalization abilities of the algorithm. This is all the more relevant given that we could only train our deep learning-based models on a relatively small set of patients, whereas there is a large inter-patient morphological variability in the ICP waveform. In the case of our testing dataset, a preselection of candidate peaks using a search for the local maxima of the curvature function improved the algorithm’s ability to discriminate pulses with a P2/P1 ratio greater than one. The observed improvements in accuracy amounted to 1% for the recurrent network and 3% for our U-Net, respectively. The MAE obtained for the estimated P2/P1 ratio was about 0.05 on both testing sets. To the best of our knowledge, this is the first time that an MAE for the P2/P1 ratio has been mentioned in the literature. However, this performance suggests that our pipeline is accurate enough to be combined with the P2/P1 interpretation grid developed by Ballestero et al. [46].

Due to its high variability, the output P2/P1 ratio signal has to be interpreted after the postprocessing step. Simplifying the raw output signal in this way implies a tradeoff between the readability and the faithfulness of the information displayed to the end user. The proposed algorithm could still be improved in various ways. For instance, the delay induced between the raw and the postprocessed P2/P1 ratio signal could be alleviated by weighting the contribution of each point to the moving average in favor of the most recent points.

The biggest limitation of our study is that only 10 patient recordings contributed to the pulse database. Because of this small number, we chose to include samples from each of the ten patients in both the training and testing datasets to train our neural networks using as much diversity as possible. By doing this, we made the assumption that the ICP signal variability for a single patient over eight days (that is to say, the average monitoring duration) was enough to neglect the effects of a common underlying distribution. However, the generalization abilities of our automated framework still need to be improved by expanding our datasets with further inclusions. This is all the more important since we obtained quite different true-positive rates during the model selection (78%) compared to the final automated framework evaluation (87.3%).

When designing the data processing pipeline, we considered taking into account the neighborhood of each single pulse better. For instance, the pulse selection process could have integrated all the pulses occurring over the last minute prior to the one being classified, thus helping interpret the pulse waveform. However, this would have required a much more computationally intensive training step since the recurrent networks would have had to capture more long-term dependencies. In addition, the database would have needed to consist of contiguous labeled samples, which could have limited the diversity achievable through manual labeling in this manner. We faced the same issue when sampling the final testing dataset, which was particularly imbalanced, with 90% of its false-negative pulses occurring in the same segment.

This observation leads us to discuss the main drawbacks of monitoring the P2/P1 ratio. As mentioned earlier, this information is not always available and depends on biological mechanisms still not fully understood [21]. As cerebral compliance can also be seen as a reserve volume, Godoy et al. suggested that the volume of contusions or edema could increase the P2/P1 ratio [19]. A more complete picture of cerebral compliance could be obtained by combining the P2/P1 ratio with other indicators such as the mean ICP, pulse amplitude [47], or pulse-shape index [45]. More generally, cerebral compliance needs to be considered as part of a bundle of information available to patients. Characterizing it is especially helpful when the ICP is close to the hypertension threshold, as a simple mean calculation is not informative enough for the current state of the cerebrospinal system. Cerebral compliance may also provide information for specific decisions, for instance, when it comes to adjusting or ending sedation.

## 6. Conclusions

Our automated detection framework allows for P2/P1 ratio monitoring on ICP ratio signals without needing any other input data. Its conception was developed under this constraint to facilitate its implementation into onboard bedside devices. Pulse selection and subpeak designation are performed using LSTM-based recurrent networks, which outperformed CNN networks in both tasks. Although a larger testing database is needed to more accurately assess the performance of the full data analysis pipeline, experiments on a 10-patient dataset achieved promising results. Monitoring the P2/P1 ratio, when possible, contributes to creating a more precise picture of the cerebrospinal system, alongside other indices such as the mean ICP or pulse amplitude.

## Figures and Tables

**Figure 1 sensors-23-07834-f001:**
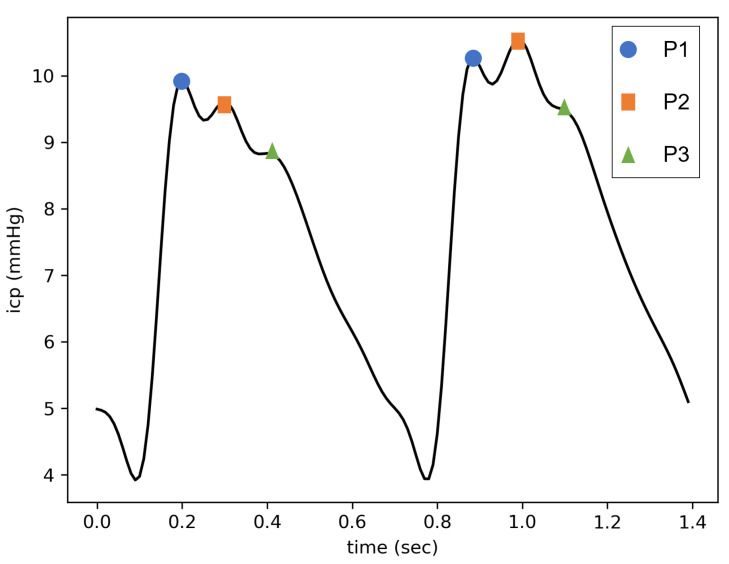
Two pulses of cardiac origin on an ICP signal. The left one has a P2/P1 ratio < 1 (i.e., it corresponds to a normal ICP pulse shape), whereas the right one has a P2/P1 ratio > 1.

**Figure 2 sensors-23-07834-f002:**
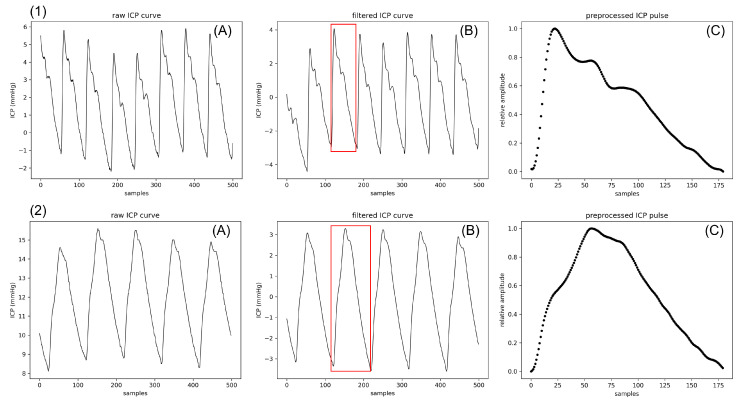
Complete preprocessing of two ICP signals from patients 1 and 2. The raw signal (**A**) is filtered between 0.3 Hz and 20 Hz (**B**). Pulses are provided to our subpeak identification pipeline after being normalized in amplitude and length. For each row, the normalized pulse displayed in (**C**) corresponds to the pulse framed in red before normalization.

**Figure 3 sensors-23-07834-f003:**
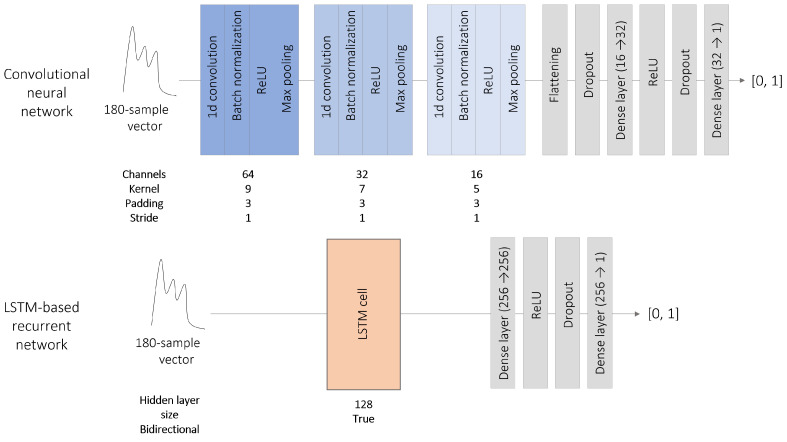
CNN and LSTM-based recurrent network architectures used for pulse selection. In both cases, a dropout with a probability of 0.2 is applied. A sigmoid function is used to map the NN output to the interval [0, 1].

**Figure 4 sensors-23-07834-f004:**
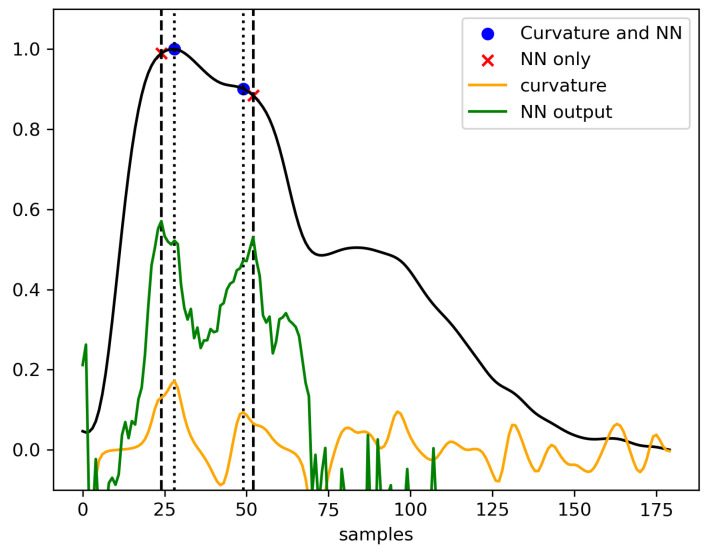
Comparison of the two methods of the peak designation algorithm. P1 and P2 are designated from among the local maxima of the curvature or directly on the basis of the NN output. The thinner dotted black lines correspond to the two selected local curvature maxima, whereas the thicker ones correspond to the two selected NN output local maxima.

**Figure 5 sensors-23-07834-f005:**
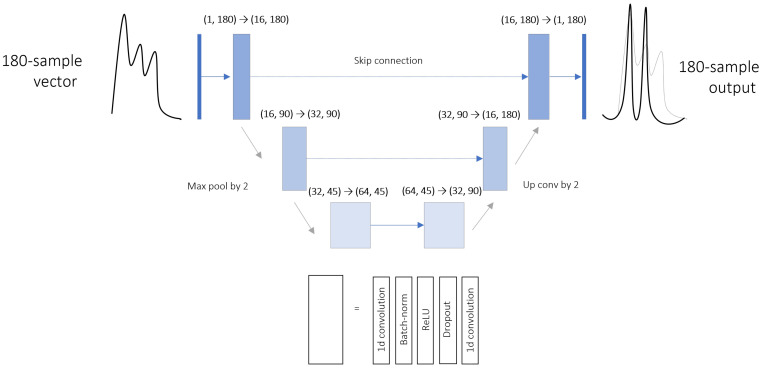
U-Net architecture proposed for subpeak detection. The NN learns to reconstitute the sum of two Gaussian curves, respectively, centered on $p_{1}$ and $p_{2}$.

**Figure 6 sensors-23-07834-f006:**
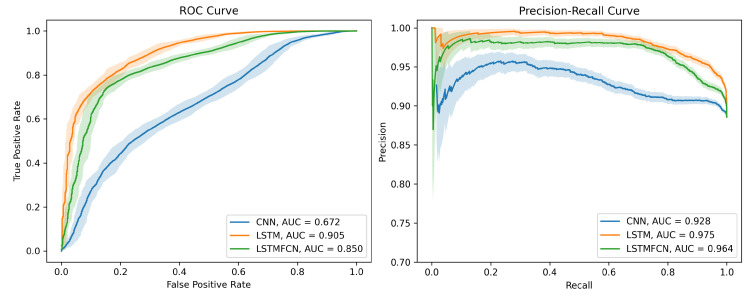
ROC (**left**) and precision–recall curves (**right**) of the three neural network architectures used for pulse selection. Positive class corresponds to pulses with a calculable P2/P1 ratio.

**Figure 7 sensors-23-07834-f007:**
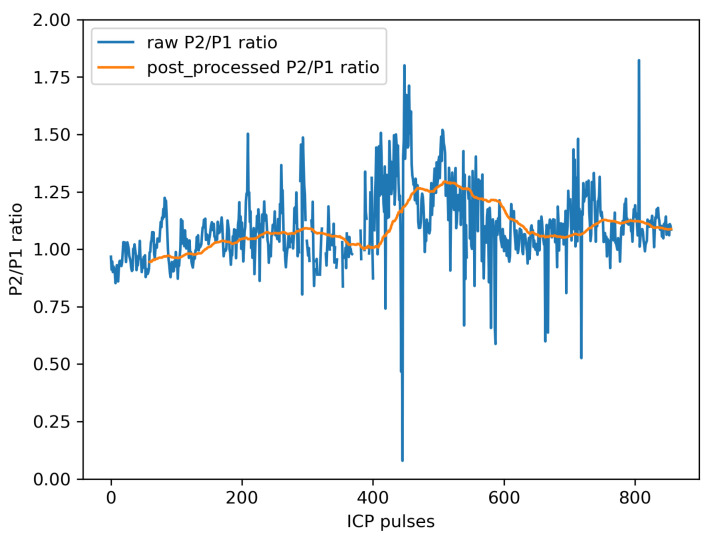
Example output for a 10-min ICP signal segment processed using the final automated framework.

**Table 1 sensors-23-07834-t001:** Confusion matrices of the 3 NN architectures compared for pulse selection at the respective optimal decision thresholds. Positive (+) class corresponds to pulses with a calculable P2/P1 ratio.

NN Architecture	CNN		LSTM		LSTM-FCN	
Prediction	−	+	−	+	−	+
True −	399	98	421	76	397	100
True +	1865	1982	847	3000	1005	2842
True-Positive Rate (%)	51.5		78.0		73.8	
False-Positive Rate (%)	19.7		15.3		20.1	

**Table 2 sensors-23-07834-t002:** Performance of five methods for P1 and P2 detection. P1 and P2 are designated as the two candidate subpeaks corresponding to the two highest NN output values. Local maxima of either the curvature or NN output are selected as candidate subpeaks. As a baseline, the algorithm “Curvature” corresponds to the designation of the two first local maxima of the pulse curvature as P1 and P2. Mean absolute errors (MAE) for the appearance times of P1 and P2 are expressed as a percentage of the total pulse duration.

Algorithm	Selection of Candidate Peaks	P1 MAE (%)	P2 MAE (%)	Ratio MAE	Accuracy (%)
1D U-Net	NN output	1.2 ± 0.1	2.1 ± 0.2	0.08 ± 0.03	93.2
	Curvature	0.6 ± 0.05	2.2 ± 0.2	0.05 ± 0.02	96.6
LSTM	NN output	0.70 ± 0.05	1.3 ± 0.07	0.03 ± 0.003	96.9
	Curvature	0.70 ± 0.06	1.3 ± 0.2	0.05 ± 0.02	97.3
Curvature	-	2.4 ± 0.2	4.0 ± 0.2	0.1 ± 0.01	89.3

**Table 3 sensors-23-07834-t003:** Performance of the final automated P2/P1 ratio computation framework. Metrics associated with P2/P1 ratio values (i.e., P2/P1 ratio MAE and accuracy of ratio > 1 detection) are only calculated onpulses with a labeled P2/P1 ratio value that passed the selection step.

True-Positive Rate (%)	False-Positive Rate (%)	P2/P1 Ratio MAE	Accuracy of Ratio > 1 Detection (%)	Ratio-Associated Pulses (%)	Displayed Ratio Time (%)
87.3 *	14.6	0.044 ± 0.002	99.7 *	85.8	88.3

* significantly higher than the same metric calculated on the testing set during NN selection (*p*-value < 0.05).

**Table 4 sensors-23-07834-t004:** Confusion matrix obtained for the final pulse selection step. Positive class corresponds to pulses with a calculable P2/P1 ratio.

	Predicted −	Predicted +
True −	499	85
True +	554	6261

## Data Availability

The data are not publicly available due to medical confidentiality.

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
