# Peer review of "A Deep Learning-Based Automated Framework for Subpeak Designation on Intracranial Pressure Signals"

_sensors, 2023, doi:10.3390/s23187834_

Round 1

Reviewer 1 Report

This manuscript utilizes advances in Machine Learning to address an important problem in the management of patients with traumatic brain injuries. In particular, it aims at automatically estimating the compliance based on the presence of the two sub-peaks within an intracranial pressure pulse.

Here are some comments:

- ICP was acquired at a frequency of 100 Hz which seems to limit the ability of the system to capture details of the ICP waveform. It would be important to show some of the raw ICP waveforms before pre-processing and confirm that the peaks can actually be seen in those. If not, it is an important point to discuss.

- Could you show the raw ICP waveform before and after pre-processing with the the third degree polynomial fitting (and other filters)?

- Section 3.3 is unclear. Start with a summary of what you are describing and follow with the details of the description.

- figure 2 and 4 are too small

- Figure 5 - in addition to ROC curves, it would be useful to show Precision/recall curves.

- Add confidence interval for AUC-ROC values

- For TPR and FPR of Table 1, report the TPR and FPR at optimal operating point of the ROC curves.

- Figure 6 is not described and not convincing. If you still want to include it, you should discuss tracking/filtering methods for ICP peaks such as the one below 

Bayesian Tracking of Intracranial Pressure Signal Morphology. Artif Intell Med, October, 2011.

Reviewer 2 Report

Intracranial pressure (ICP) is classically monitored invasively in intensive care units (ICU) in the event of brain injury. One of the main objectives for a clinician is to attempt to maintain the intracranial pressure at a reasonable range to limit potential for intracranial injury.  Intracranial compliance or the ability of the brain to accommodate changes in intracranial pressure is a critical issue.  Heartbeat-induced pulses in ICP varies according to cerebral compliance.  Classification of these pulses can be important in bedside monitoring of patients with intracranial pressure issues.  Cerebral compliance must be considered as part of the information available on patients.  Characterizing it is especially helpful when ICP is close to the hypertension threshold, as a simple mean calculation is not informative enough on the current state of the cerebrospinal system.  Cerebral compliance may also provide information for specific decisions, for instance when it comes to adjusting or putting sedation to an end.  Better understanding of these pulses may be helpful in the bedside monitoring of patients with intracranial hypertension problems.  However, the clinical applications of this work are probably somewhat limited.

Minor improvements in the quality of the English could be made.

Reviewer 3 Report

The paper written by the following Authors: Donatien LEGÉ, Laurent GERGELÉ, Marion PRUD’HOMME, Jean-Christophe LAPAYRE, Yoann LAUNEY and Julien HENRIET, entitled “A Deep Learning-Based Automated Framework for Subpeak Designation on Intracranial Pressure Signals” presents an interesting study on intracranial pressure signal monitoring with deep learning technique.

Although the paper is interesting, I have some major concerns:

Title

The title reflects the results presented here.

Abstract

The abstract is lacking the aim of the study, material and methods description as well as an informative conclusion. It should be written in more details.

Introduction

In the introduction part Authors should add some overall information in paragraph/paragraphs dedicated to artificial systems applied in blood flow simulation.

- A Novel Patient-Specific Human Cardiovascular System Phantom (HCSP) for Reconstructions of Pulsatile Blood Hemodynamic Inside Abdominal Aortic Aneurysm, 10.1109/ACCESS.2018.2876377

Material and Methods

1. Boundary conditions and initial conditions should be described in more details.

Results

1. How medical tools applied for the medical diagnostics may influence preoperation prognosis?

2. Authors indicated “The number of false-positive pulses and false-negative pulses correspond to 1.8% and 9.7%, respectively, of the total testing data set when using the LSTM-based architecture for classification. In contrast, these percentages amount to 2.3% and 42.9%, respectively, when using the convolutive network.”. The practical aspect should be presented.

Discussion

1. More literature should be applied in the discussion part. Currently Authors included only 4 papers.

Round 2

Reviewer 3 Report

Unfortunately, the Authors did not respond to my comments. For example, adding numbers (new literature) to the current discussion without supplementing the text with new information cannot be called a discussion of the results.

Adding numbers to the existing manuscript text is not a part of the discussion. The authors were asked to refer their results to the literature in the same area. The authors did not do it.  Authors only added numbers, which indicates that they do not understand what the discussion of the results obtained is.

Considering the above, and the fact that most of my comments were not addressed, I believe that the manuscript does not meet the basic requirements for publication and should be rejected.

Author Response

Unfortunately, the Authors did not respond to my comments. For example, adding numbers (new literature) to the current discussion without supplementing the text with new information cannot be called a discussion of the results.

The results in the Discussion are now compared in the Discussion with results obtained in the literature:

- “The performances obtained for the chosen threshold correspond...”

-”The MAE obtained on the estimated P2/P1 ratio…”

Adding numbers to the existing manuscript text is not a part of the discussion. The authors were asked to refer their results to the literature in the same area. The authors did not do it.  Authors only added numbers, which indicates that they do not understand what the discussion of the results obtained is.

Considering the above, and the fact that most of my comments were not addressed, I believe that the manuscript does not meet the basic requirements for publication and should be rejected.

In the first review cycle, the authors mentioned that most of the reviewer’s comments did not seem to be relevant to the subject of this article, and that the reviewer’s expectations probably needed to be clarified. If the reviewer can explain the link between blood flow simulation and the detection of subpeaks on a pressure signal, the authors will be pleased to consider the aforementioned comments.